# OpenReview forum: "BrokenMath: A Benchmark for Sycophancy in Theorem Proving with LLMs"
_ICLR.cc/2026/Conference — Submitted to ICLR 2026_

### Official Review · Reviewer_bA42 · 2025-10-26

**Soundness:** 3
**Presentation:** 3
**Contribution:** 3
**Rating:** 6
**Confidence:** 3

**Summary:**

This work finds that the LLMs are prone to hallucination and sophistry for theorem proving. Therefore, this work builds a benchmark named BrokenMath to evaluate the sycophantic behavior in LLMs in the natural language theorem proving.

**Strengths:**

* This problem is interesting. This work proposes to evaluate the hallucination and sycophancy, focusing on theorem proving.
* The dataset construction is clear. The figure 1 present the dataset construction pipeline.
* This work discuss several interesting observation, such as self-sycophancy.
* The experiment parts contain various models, suggesting that the sycophancy is common in current SOTA models.
* The prompt is given in Appendix to help reproduce the resutls.

**Weaknesses:**

* The evaluation uses LLM. Therefore, it is not clear whether the evaluation is correct
* Qwen3-4B sycophancy is even lower than Qwen3-235B and DS-V3.1. The question is why the smaller model has a lower sycophancy rate?
* Figure 7 presents that the model performance is significantly related to the prompt, which suggests that the evaluation variance may be large with different methods.

**Questions:**

N/A

---

> ### Author Response · Authors · 2025-11-19
> **Rebuttal**
>
> We thank the reviewer for their constructive feedback and for finding the topic interesting and our work thorough, clearly presented, and insightful. We address their remaining concerns below.
>
> **How do you ensure the use of an LLM-as-a-judge does not impact correctness?**
>
> We initially conducted a thorough validation of our LLM-as-a-judge system, on which GPT-5-mini achieved 95% accuracy on our granular 4-way classification task. We have now conducted further analysis, which is detailed in the common response (Q.1). Our key findings are:
> - GPT-5-mini achieves a near-perfect **97.9%** binary classification accuracy between sycophant and non-sycophant problems.
> - A new evaluation using Grok-4-Fast as a judge confirmed that there is no measurable bias for or against models from the judge's own family.
>
> These results give us high confidence that our LLM-as-a-judge provides a reliable and unbiased evaluation for this task.
>
> **Why do smaller models, like Qwen3-4B have lower sycophancy than larger ones, such as Qwen3-235B?**
>
> We believe there are two main factors behind this. First, current models are trained to always provide an answer, as both benchmarks and reinforcement learning techniques tend to reward only correct responses without penalizing incorrect guesses. As a result, “bigger” does not necessarily mean “better” on our benchmark. In fact, Qwen3-235B is likely more capable of producing plausible guesses, which may actually worsen its performance on sycophancy-related tasks.
>
> Second, after manually reviewing a small subset of samples, we observed a consistent behavioral difference between the models. Qwen3-4B more frequently “brute-forces” problems: trying small cases, performing explicit calculations, and probing the structure of the question. This trial-and-error approach naturally leads to counterexamples in particular types of problems. By contrast, Qwen3-235B relies far less on these concrete computations, making it less likely to discover such counterexamples.
>
>
> **What do the results in Figure 7 imply about the evaluation variance?**
>
> This experiment presented in Figure 7 is unrelated to evaluation variance. Concretely, this experiment used a specific mitigation prompt designed to make the LLM aware of the possibility of an incorrect problem statement. The resulting variability across models demonstrates that this strategy has varied effectiveness across different models, and as such does not say anything about evaluation variance itself.
>
> To address the general concern of statistical uncertainty, we have now included 95% confidence intervals (typically around ±4%) for all main experiments in the paper.
>
> Further, as detailed in our common response (Q.2), we tested additional targeted prompting techniques. None were able to fully eliminate the sycophantic behavior, reinforcing our paper's conclusion. We conclude that while prompt engineering is a significant factor, it is likely insufficient on its own to solve the problem of sycophancy in mathematical reasoning.

---

### Official Review · Reviewer_FxiE · 2025-10-31

**Soundness:** 1
**Presentation:** 3
**Contribution:** 2
**Rating:** 2
**Confidence:** 4

**Summary:**

This paper introduces BrokenMath, presented as "the first benchmark for evaluating sycophantic behavior in LLMs within the context of natural language theorem proving." The benchmark contains 504 problems from 2025 mathematical competitions, where each problem is perturbed to produce a false statement. The authors evaluate 10 state-of-the-art LLMs and find that even the best model (GPT-5) exhibits "sycophantic" behavior 29% of the time. The paper investigates factors influencing this behavior (difficulty, problem type) and evaluates mitigation strategies including prompt engineering and fine-tuning.

**Strengths:**

1. **High-quality data curation**: The benchmark uses recent 2025 competition problems with expert verification, reducing contamination risks compared to prior work using GSM8k or AIME.

2. **Comprehensive evaluation**: Systematic assessment of 10 frontier models across multiple dimensions, providing valuable empirical comparisons.

3. **Fine-grained behavioral classification**: The four-category taxonomy (Ideal/Corrected/Detected/Sycophant) provides more nuanced information than binary correct/incorrect classification.

4. **Thorough experimental execution**: Careful validation of LLM-as-a-judge (95% agreement with human labels), detailed ablation studies, and multiple experimental settings.

**Weaknesses:**

### 1. **Fundamental construct validity failure: Confounding competence with alignment**

The paper claims to measure "sycophancy" (an alignment issue) but provides no mechanism to distinguish it from mathematical incompetence. True sycophancy requires that a model **can** judge a statement's validity but chooses not to due to user-pleasing tendencies. The paper's methodology cannot differentiate:

- **Phenomenon A**: Model accepts false statement because it cannot judge truth/falsity (competence deficit)
- **Phenomenon B**: Model can judge but suppresses critical thinking to please user (alignment sycophancy)

A rigorous operationalization would require:
```
For proposition P and its negation ¬P:
  Step 1: Filter to problems where model can correctly prove P
  Step 2: Further filter to problems where model can correctly disprove ¬P
  Step 3: Only on this filtered set, if the model still attempts to prove ¬P without questioning or refuting it, does this represent sycophancy.
```

The paper performs no such filtering, rendering all claims about "sycophancy as an alignment problem" unsubstantiated.

### 2. **Table 2 reveals that the measurement is fundamentally contaminated**

Table 2 presents "sycophantic behavior" split by whether models can solve the original problem:
- Solved problems: 21.5% (GPT-5 example)
- Unsolved problems: 47.7% (GPT-5 example)
- Gap: 26.2 percentage points

The existence of this table itself demonstrates the methodological failure. The "sycophancy" measurements on unsolved problems are **not measuring sycophancy at all**—they are measuring the model's inability to judge mathematical validity. This is pure competence deficit misclassified as alignment failure.

The solved/unsolved gap does not reveal "difficulty as an influencing factor." It reveals that the reported sycophancy rates are **contaminated metrics** mixing:
- True potential sycophancy (from solved subset, and even this requires further validation per Step 2 above)
- Competence limitations (from unsolved subset, which should not be in the dataset at all)

There should be no "All/Solved/Unsolved" breakdown because only the subset satisfying both Steps 1 and 2 should be included in the benchmark. The presence of unsolved problems in the evaluation fundamentally invalidates the sycophancy measurements.

### 3. **Cascading invalidation of all derivative analyses**

With the core measurement conflating competence and alignment, all subsequent analyses become uninterpretable within the paper's claimed framework:

- **Main results (§4.1)**: The reported rates (29%-70%) are inflated by competence limitations, cannot quantify alignment issues
- **Difficulty analysis (§4.2)**: The paper interprets the solved/unsolved gap as "higher difficulty implies higher sycophancy," but this correlation precisely demonstrates that the measurement captures the mixture of alignment and competence rather than only alignment
- **Problem type comparison (§4.2, Fig 4)**: Cannot distinguish whether proof-style vs. final-answer differences reflect difficulty (competence) or alignment dynamics
- **Self-sycophancy (§4.3)**: Cannot determine if increased rates reflect consistency bias (competence-related) or alignment failures
- **Mitigation strategies (§5)**: The "modest" improvements are uninterpretable—are they trying to enhance critical reasoning ability or adjust alignment? The experimental design cannot answer this

### 4. **Mischaracterization of related work**

The paper states that prior works "typically modify existing final-answer problems... Results consistently show that frontier models are prone to sycophancy" (§2, citing Xue et al., Kirichenko et al., Liu et al., Sun et al., Ouyang, Rahman et al., Ma et al.). But indeed none of these seven papers use "sycophancy" as their core framework.

### 5. **Method is isomorphic to prior work despite claims of novelty**

Despite criticizing prior work for using "ill-posed questions" versus "well-posed but false" statements (§1), the perturbation approaches are structurally identical:
- Prior work: Perturb problems (remove constraints/add contradictions) → test detection
- This work: Perturb problems (change conclusions) → test detection

Both evaluate the same capability: detecting problematic mathematical inputs. The "ill-posed" vs. "well-posed but false" distinction does not constitute methodological innovation—both test critical reasoning.

**Questions:**

The main questions and concerns are detailed in the Weaknesses section above.

---

> ### Author Response · Authors · 2025-11-19
> **Rebuttal (1/2)**
>
> We thank the reviewer for their detailed feedback and for acknowledging the high quality of our data curation, the comprehensive and nuanced nature of our evaluation methodology, and the thoroughness of our experimental execution. We believe there may be a disagreement regarding the definition of sycophancy and its implications, which we aim to clarify below. We welcome the opportunity to discuss these points further.
>
> **Can you safely claim that the results of your benchmark clearly represent sycophantic behavior?**
>
> We thank the reviewer for this critical question, which touches on the core definitional basis of our work. We respectfully disagree with the alternative definition proposed by the reviewer for three main reasons: (1) our definition of sycophancy is widely-accepted in recent literature, (2) we have conducted a new experiment that directly demonstrates the observed behavior is a prompt-dependent bias, not simply incompetence, and (3) our definition is more in-line with practical scenarios.
>
> 1. Recent work [1, 2, 3, 4] defines sycophancy as the tendency of a model to align its output with a user's stated desire, even when that desire is factually incorrect. Thus, these papers solely observe whether the model's stated belief changes based on the user's framing and do not take into account model capability. Our work applies this same principle to the domain of mathematical reasoning.
>
> 2. To directly address the reviewer's concern and empirically test whether the behavior is mere incompetence or prompt-driven, we conducted a new experiment. For a problem $P’$ in our benchmark, we created prompts asking the model to prove the converse $\neg P’$, which is a factually correct task.  In 97.7% of cases, one of the weaker models, Qwen3-4B, correctly followed the instruction and attempted to disprove $P’$. We manually went through the remaining 2.3%, finding that all errors stemmed from misunderstanding the problem. This demonstrates that the model's behavior is overwhelmingly dictated by the prompt's instruction. When told to prove the false $P’$, it tries to do so; when told to prove the true $\neg P’$, it also does. This is clear evidence of prompt-dependent behavior and therefore sycophancy, not a fixed competence deficit. We have included this analysis in App A.2.
>
> 3. Real-world users will not pre-filter their queries based on a model’s unknown capabilities. Therefore, they need to know how rigorously to verify outputs when the model’s competence is uncertain. Under our definition, GPT-5’s 30% score clearly indicates that users should apply very thorough verification. In contrast, under the reviewer’s proposed alternative definition, the benchmark results offer little practical value for these users.
>
> Finally, we manually examined reasoning traces of open models and found that, in the majority of cases, the models recognize that something is wrong but fail to communicate this to the user. Relevant examples are provided in Appendix E. Importantly, this behavior may occur more frequently than the utility metric indicates: an adjusted problem $P'$ derived from an original problem $P$ must satisfy $P \rightarrow \neg P'$. However, the inverse implication ($\neg P' \rightarrow P$) does not necessarily hold, making $\neg P'$ often easier to prove than $P$.
>
> **Does Table 2 reveal that the measurement is contaminated?**
>
> No, we argue this table reveals a crucial finding of our paper: sycophantic tendencies are significantly amplified by a model's lack of competence. A model is more likely to be deceptively agreeable when it is on shaky ground. As mentioned above, models almost always (attempt to) prove $\neg P’$ when asked to do so, clearly indicating that no matter whether it is in their capabilities or not, they do change their outputs based on user input.
>
> **Can the authors redefine their characterization of prior work?**
>
> Yes, we have updated our related work section to more accurately frame this prior research as focused on the broader areas of "reliability and robustness of LLMs in mathematics."
>
> (continued in 2nd response)

---

> ### Author Response · Authors · 2025-11-19
> **Rebuttal (2/2)**
>
> **Is the methodology behind BrokenMath fundamentally novel?**
>
> We clarify the key methodological advancements of BrokenMath over prior work, such as [5]. While the high-level strategy "perturb-then-test" can be applied broadly, even for many works in the field of general LLM robustness, it overlooks the crucial distinctions in data quality, realism, and evaluation that support our contribution:
>
> 1. **Minimal contamination:** We use recent problems from sources released after model training cutoffs. As shown in App A.1f, we have now performed an additional experiment to confirm that contamination is indeed minimal.
> 2. **Rigorous human-in-the-loop validation**: Every problem was verified by a human expert to ensure the perturbed statement is both incorrect and closely resembles the original problem structure.
> 3. **Nuanced problem types:** Our methodology is applicable, regardless of the problem type, unlike [5], which requires the existence of a final answer.
> 4. **Robust evaluation:** Prior work [5] relies on unverified, parser-based methods (e.g., checking for hard-coded abstention strings) which can easily fail due to formatting mistakes. Our LLM-as-a-judge framework is far more robust, evaluating the semantic intent of the response. As validated in our paper and the common response (Q.1), this framework achieves near-perfect human agreement.
>
> Therefore, the end product of our work, BrokenMath, is a necessary and significant improvement over prior benchmarks that enables the detailed analysis we make in our paper.
>
> ### References
> [1] https://arxiv.org/abs/2310.13548
>
> [2] https://arxiv.org/abs/2311.09410
>
> [3] https://arxiv.org/abs/2312.09085
>
> [4] https://arxiv.org/abs/2505.13995
>
> [5] https://arxiv.org/abs/2507.03133v1

---

### Official Review · Reviewer_XyWR · 2025-11-01

**Soundness:** 2
**Presentation:** 3
**Contribution:** 3
**Rating:** 6
**Confidence:** 4

**Summary:**

The paper introduces BROKENMATH, a benchmark to measure sycophancy - LLMs "going along" with incorrect prompts - in natural-language theorem proving. Authors build 504 problems by perturbing recent (2025) competition problems into false but plausible statements with LLM assistance and expert verification, then evaluate frontier models using an LLM-as-a-judge protocol. They find sycophancy is widespread and worse on harder/proof-style tasks; several mitigations (prompting, agentic variants, fine-tuning) help but don’t eliminate it.

**Strengths:**

1. Interesting and original approach that moves beyond final-answer math to proof-style tasks with verified false statements, addressing contamination and ill-posedness critiques of prior datasets.

2. Studies self-sycophancy and agentic sycophancy (best-of-n, iterative verification), expanding the phenomenon’s scope.

3. Careful comparisons across problem types and difficulties; shows sycophancy rises on problems a model cannot solve.

4. Empirically important finding: even top models are sycophantic a non-trivial fraction of the time (29% for GPT-5), especially on proofs, helping recalibrate expectations for theorem-proving deployments.

**Weaknesses:**

1. The main classification relies on an LLM-as-a-judge. Although a 95% agreement is claimed, the paper would benefit from a larger human-labeled audit, inter-annotator agreement, and/or error analysis (e.g., where the judge mistakes “Detected” vs “Corrected”). Also, the judge choice could correlate with family-level behaviors and subtly advantage similar model families.

2. Perturbations are LLM-generated then expert-tuned; there’s a risk that models learn to spot stylistic artifacts of the perturbation procedure.

3. Focuses on advanced high-school/undergrad level problems; unclear generalization to research-level math or to formal-proof ecosystems. Authors note this, but it constrains impact.

4. While the dataset is constructed to span algebra, geometry, combinatorics, and number theory, the results don’t report per-domain sycophancy/utility or domain-specific failure modes. Given well-known differences in LLM math behavior (e.g., geometry often needs diagrammatic or synthetic reasoning; number theory leans on modular arguments), a topic-level analysis could surface systematic vulnerabilities and make the benchmark more diagnostic.

**Questions:**

1. Can you report more details on the bracket/judge prompts for best-of-n, stopping criteria for iterative verification, and cost/latency, or did I miss them somewhere?

2. The mitigation evaluation depth can be potentially extended to cover broader prompt families.

---

> ### Author Response · Authors · 2025-11-19
> **Rebuttal**
>
> We thank the reviewer for their feedback and for finding our approach interesting, original, and empirically relevant. We address their remaining concerns and questions below.
>
> **Can the authors provide further error analysis on the LLM-as-a-judge evaluation?**
>
> Yes, we added expanded analysis in Q.1 of the common response and a new section in the paper, App A.4, including:
>
> - Conducted a cross-model bias analysis, which shows no significant bias when using a judge from a different model family (Grok-4-Fast).
> - Performed a qualitative error analysis of the few failure cases from our human validation, confirming the judge's reliability on the core sycophancy detection task.
>
> **Can LLMs potentially spot stylistic artifacts of the perturbation procedure?**
>
> No, this is very unlikely. We have taken several precautions to mitigate this problem:
>
> 1. Our curation process, which involved expert human review, was designed to ensure the resulting false statements were semantically plausible and flowed naturally. This makes it unlikely that the problems are distinguishable from genuine competition tasks, especially for the proof-based questions where the style is nearly identical to the originals.
>
> 2. As is standard practice for LLM benchmarks, we operate under the principle that evaluation datasets should not be used for model training. This prevents models from overfitting to any potential stylistic artifacts present in BrokenMath.
>
> **Can the methodology generalize to research-level mathematics or formal-proof systems?**
> Yes, we believe the core methodology can be adapted with the following considerations:
> - Research Mathematics: Our pipeline is designed to be general and could readily be applied to statements from research papers. The primary challenge is collating a high-quality set of plausible-but-false research-level conjectures, which requires substantial domain-specific effort from PhD-level mathematicians. We view this as an exciting and important direction for future work.
> - Formal Systems: The phenomenon of sycophancy is inherently mitigated in formal systems. While a model could be prompted to attempt to prove a false statement, the formal verifier would simply reject any invalid proof. Therefore, while models might still exhibit biased behavior, these systems prevent it from propagating into the final outcome.
>
> **Can the authors provide domain-specific failures or utility/sycophancy scores?**
> Yes, we have performed this domain-specific analysis and included the full results in Appendix A.5. Here are the key findings:
>  - Utility: Most models perform best on Algebra, likely due to the domain's standardized symbolic algebraic techniques. Performance is more uniform across the rest, with some models showing specific strengths (e.g., Gemini-2.5-Pro in Number Theory).
>  - Sycophancy: Models are significantly less sycophantic on Algebra and Number Theory. We hypothesize this is because these problems often have counterexamples that can be found via direct, symbolic exploration. Conversely, Geometry and Combinatorics problems induce higher sycophancy rates, as they may require more creative or visual reasoning.
>
> A particularly interesting failure mode appeared in Geometry, where models would begin an algebraic approach, suspect an error in the problem statement, but then discard their doubt to hallucinate a faulty "synthetic" (purely geometric) proof. We have included an example in Appendix E.2.
>
> **Can you report more details on the bracket/judge prompts for best-of-n, stopping criteria for iterative verification, and cost/latency?**
>
> Yes, we have included a small description of both agents in Appendix B.3, and the bracket prompt in Appendix D.10. We did not make any changes to the agents, directly implementing them from prior work and we therefore refer the reviewer to these papers for full details.
>
> Regarding the stopping criterion for iterative reflection: the agent either stops when (1) it has done 4 iterations, (2) the verifier accepted the provided proof, or (3) the verifier rejected the provided proof 3 iterations in a row. These parameters are lower than the parameters by the related work due to cost considerations.
>
> Unfortunately, we did not measure latency, and since we ran these agents on local machines, we cannot provide any monetary cost. However, we can report that for the best-of-n agent, output tokens increased by an average of 4.8, and for the verification agent by a factor of 5.1.
>
> **Can you provide a more in-depth evaluation of different families of mitigation prompts?**
>
> Yes, we have now performed an evaluation on two additional styles of mitigation: a two-step instruction where the model first seeks a counterexample, and one where the model is strongly incentivized against conformity. Both yielded worse results than our original mitigation prompt on Qwen3-4B, with a 44.6% and 49.4% sycophancy rate respectively  We have provided a more thorough description in **Q.2** of the common response.

---

> > ### Comment · Reviewer_XyWR · 2025-11-25
> >
> > Thanks for the responses. I would like to maintain my positive scores.

---

### Official Review · Reviewer_evjq · 2025-11-05

**Soundness:** 3
**Presentation:** 3
**Contribution:** 3
**Rating:** 6
**Confidence:** 4

**Summary:**

The paper introduces the first benchmark for evaluating sycophancy in natural-language theorem proving. The benchmark uses challenging 2025 math olympiad problems to minimize contamination, generating subtle false statements via an LLM and verifying them with experts. The dataset has 504 examples (321 proof, 183 final-answer). Evaluation of frontier models shows pervasive sycophancy, especially in proof tasks and with increasing difficulty. Existing mitigation techniques provide only limited benefit.

**Strengths:**

1. Important and timely focus on high-stakes alignment failure in mathematical reasoning.
2. Rigorous counterfactual construction: LLM perturbation + expert verification → plausible and difficult false theorems.
3. Clear exposition; strong motivation and methodology description.
4. Empirical results provide useful diagnostic insights: proof settings and harder tasks produce more sycophancy.

**Weaknesses:**

1. LLM-as-judge introduces circularity and evaluation risk; the judge may share the same biases.
2. Dataset size (504 samples) limits robustness and granularity in difficulty-stratified analysis.
3. Assumption of minimal contamination relies on recency; no quantitative verification.
4. Limited mechanistic analysis of how proofs fail (e.g., where sycophancy manifests in reasoning chains).

**Questions:**

1. Can you provide quantitative evidence (e.g., perplexity checks, memory probing) supporting minimal pre-training contamination?
2. What is the human vs. judge agreement rate on incorrect proofs where sycophancy is subtle?

---

> ### Author Response · Authors · 2025-11-19
> **Rebuttal**
>
> We thank the reviewer for their feedback and acknowledging our work as important and timely, and our analysis as providing useful diagnostic insights. We address any remaining questions and concerns below.
>
>
> **Does the LLM-as-judge introduce circularity and evaluation risk?**
>
> To empirically test for such biases, we conducted a new evaluation using Grok-4-Fast, a capable model from a different model family, as an out-of-family judge. We provide the full results table and a more detailed error analysis in **Q.1** of the common response.
>
> Our key finding is that the results are highly consistent across both judges, with negligible bias (<0.5%) observed between GPT-5-mini and Grok-4-Fast. This demonstrates the robustness of our evaluation method against model-specific biases. We further believe that circular biases are unlikely due to the simplicity of the task, resulting in the very high performance of the judge.
>
>
> **Does the dataset size limit robustness in granularity in difficulty-stratified analysis?**
>
> We have added 95% confidence intervals to all graphs and tables in our paper. These intervals are approximately 4.5% for the main results and ~10% for lower-sample analyses. We note that this level of statistical significance is consistent with other high-quality benchmarks for frontier models (e.g., the AIME 2024/5 benchmarks use only 30 problems each) which nevertheless provide a strong signal for model capabilities. All our conclusions remain valid when taking these confidence intervals into account
>
>
> **Can the authors provide quantitative evidence supporting the claim of minimal contamination?**
>
> Yes. We conducted a quantitative analysis to estimate data contamination. Following the methodology of [1], we used a masked word prediction task on a subset of 60 problems containing uncommon names or nouns. Across the 60 problems and all models tested, models failed to correctly guess the masked word in all but one single problem, strongly indicating that the majority of the dataset is not memorized.
>
> For the single case where the name was correctly guessed (allrussian_2025_1), the guess was made by Grok-4-Fast and o4-mini. Interestingly, o4-mini was released before the problem itself was published. This suggests it may be an outlier of the probing method itself. We tried to find whether a variation of the problem might have been published prior to the competition, but could not find any evidence of this.
>
> We considered several other methods for data contamination detection, but found none to be effective. For instance, not only are logit-based methods infeasible for closed models, they have also been shown to work very poorly [2]. We also attempted the quiz-style continuation guessing method from [3], where a model is given the start of a problem and is asked to select which of 2 possible continuations is the one that appears in the actual problem. We made several attempts to make these alternative continuations sound as natural as possible, but a blind test by one of the authors on a small subset still gave an 80% accuracy, since most rephrasings used slightly unnatural synonyms for certain words.
>
> We have included this analysis in App A.1.
>
>
> **Can you investigate how sycophancy manifests mechanistically?**
>
> Of course! We have now examined the reasoning traces of several open-source models, including Qwen3-4B and GPT-OSS 120B. This analysis reveals 2 consistent patterns:
>  - The model shows initial suspicion about the problem statement but fails to find a definitive disproof. At the end of its reasoning process, it abandons its doubts and conforms to the user's prompt by inventing a flawed solution.
>  - The model finds a clear counterexample but, instead of reporting it, silently introduces an unstated assumption that makes the original flawed statement provable. The counterexample is never mentioned in the final output.
>
> Additionally, in rare cases, the model is clearly biased by the user's prompt from the start. We have included relevant examples demonstrating the 2 main behaviors in Appendix E.
>
> **What is the human vs. judge agreement rate on sycophantic proofs?**
>
> Our human validation set shows a near-perfect agreement rate for identifying sycophantic proofs, with 97.7% precision and 98.4% recall.
>
> This high agreement confirms the reliability of the LLM judge for this task. As detailed in our common reply **Q.1**, the few rare disagreements typically occur in ambiguous edge cases where a model disproves a trivial aspect of a statement while still incorrectly affirming the user's general-case premise.
>
> While the reviewer mentions the possibility of “subtle sycophancy”, we have found that such cases do not exist due to the nature of our setup: the models either attempt to prove the incorrect problem statement or clearly state their suspicions in the response.
>
> ### References
>
> [1] https://arxiv.org/abs/2503.12072
>
> [2] https://arxiv.org/abs/2406.16201
>
> [3] https://arxiv.org/abs/2311.06233

---

### Author Response · Authors · 2025-11-19
**Rebuttal Common Response**

$\newcommand{\evjq}{\textcolor{red}{evjq}}$
$\newcommand{\xywr}{\textcolor{blue}{XyWR}}$
$\newcommand{\fxie}{\textcolor{green}{FxiE}}$
$\newcommand{\ba}{\textcolor{purple}{bA42}}$


We thank all reviewers for their feedback, and we are happy to hear they find our work novel ($\xywr$, $\ba$), intriguing and important ($\evjq$, $\xywr$, $\ba$), our methodology well-constructed and well-argumented ($\evjq$, $\fxie$, $\ba$), and our evaluation thorough and impactful ($\evjq$, $\xywr$, $\fxie$, $\ba$). We now address common reviewer concerns and questions.

**Q.1 Can the authors provide additional details about the LLM-as-a-judge validation, such as a more thorough error analysis and addressing potential biases? ($\evjq$, $\xywr$, $\ba$)**

Of course! First, we want to clarify that the task for the LLM judge is very simple: it only needs to identify the type of behavior displayed by the model’s answer, which is independent of the answer’s logical correctness. The very high agreement rate between our judge and a human annotator (95%) demonstrates that the LLM judge is reliable for our evaluation. We now further validated this through two additional analyses: (1) an error analysis of the judge’s classifications, and (2) a cross-family evaluation to test for judge-specific biases.

1. Error Analysis (App A.4)
We have included a confusion matrix among the four classes in Figure 9.  Three main conclusions can be drawn from this matrix:
    - The off-diagonal entries are very small, indicating that the judge achieves high overall accuracy.
    - For our primary task (distinguishing between sycophantic and non-sycophantic answers) the accuracy increases to 98%, as most misclassifications occur within the three non-sycophantic categories. The remaining 2% come from highly ambiguous model outputs, such as answers that point out an error in the theorem only in an edge case while still attempting to provide a proof for all other cases.
    - The few remaining errors typically occur between the “Correct” and “Detected” categories, where the human judge determined that the theorem has been only partially recovered, while the LLM judge classified the answer as fully “Correct.”

2. Judge-specific biases (App A.4)
To address concerns about potential self-enhancement or model-family-specific biases (e.g., an OpenAI judge favoring OpenAI solvers), we conducted a new experiment. We re-ran our evaluation using Grok-4-Fast as the judge, which achieves a 96% accuracy when differentiating sycophantic and non-sycophantic answers, slightly lower than our current judge that has a 98% accuracy.
The results, presented below, show a negligible difference between the ratings from the 2 judges.

| Judge / Solver                  | **GPT-5 (high)** | **o4-mini (high)** | **Grok 4 Fast (Reasoning)** | **Grok 4** |
| ------------------------------- | ---------------- | ------------------ | --------------------------- | ---------- |
| **GPT-5-mini (medium)**         | $29.0 \pm 4.0 \%$           | $46.6 \pm 4.4 \%$             | $40.0 \pm 4.4 \%$                      | $34.8 \pm 4.2 \%$     |
| **Grok 4 Fast (Reasoning)**     | $29.2 \pm 4.0 \% $          | $46.4 \pm 4.4 \% $            | $40.4 \pm 4.4 \%$                      | $35.0 \pm 4.2 \%$     |
| **Bias (Grok4Fast − GPT5mini)** | **+0.2%**       | **−0.2%**         | **+0.4%**                  | **+0.1%** |

The minimal bias across all models strongly indicates that our evaluation is robust and not influenced by the judge's model family. We have updated our paper with a new section (App A.2) discussing these findings.

**Q.2 Can the authors evaluate how much different prompts may affect the sycophantic behaviour? ($\xywr$, $\ba$)**

We implemented and tested two additional prompting strategies designed to mitigate sycophancy:
 - Contradiction Search: We instruct the model to first find a counterexample before attempting to prove the statement.
 - Sycophancy Awareness: We explicitly make the model aware of its sycophantic tendencies and instruct it to avoid confirming the user's premise if it is uncertain.

The results on the Qwen3-4B model are presented below.

|                 | No mitigation | Standard mitigation | Contradiction | Aware |
| ------------------------------- | ---------------- | ------------------ | --------------------------- | ---------- |
| **Qwen3-4B**         | $55.6 \pm 4.4\%$          | $43.8 \pm 4.4\%$             | $44.6 \pm 4.4\%$                  | $49.4 \pm 4.5 %$    |

As shown, neither of these approaches improve upon our original mitigation strategy. This reinforces our paper's conclusion that while prompt engineering can provide some alleviation, it is insufficient to fully resolve sycophantic behavior in mathematical reasoning.

We have included the new prompts in Appendix D and added the analysis in App A.7.

---

### Author Response · Authors · 2025-12-01
**Executive Summary for new AC**

$\newcommand{\e}{\textcolor{red}{evjq}}$
$\newcommand{\x}{\textcolor{blue}{XyWR}}$
$\newcommand{\f}{\textcolor{green}{FxiE}}$
$\newcommand{\b}{\textcolor{purple}{bA42}}$

We thank the new AC for taking over this submission. We provide this summary to facilitate your evaluation of our submission given the unusual circumstances.

## Review Summary

### Scores: 6,6,2,6 Confidence: 4,4,4,3

Overall, most reviewers found our contribution valuable, and all had strong positive comments, finding our work novel ($\x$, $\b$), intriguing and important ($\e$, $\x$, $\b$), our methodology well-constructed and well-argued ($\e$, $\f$, $\b$), and our evaluation thorough and impactful ($\e$, $\x$, $\f$, $\b$).

# Main Concerns

There were two main concerns raised by reviewers.

**1. Errors and Biases of LLM-as-a-judge validation ($\e$, $\x$, $\b$)**
Several reviewers pointed out that the use of an LLM-as-a-judge for evaluation might lead to downstream inaccuracies. In response, we conclusively showed that errors and biases in our LLM-as-a-judge are minimal/non-existent:
- The judge has a *98% accuracy* on the binary classification used for sycophancy measurement.
- We compared GPT‑5‑mini to Grok-4-Fast as a judge, showing our results are consistent to <0.5% margin.

**2. Definition of sycophancy ($\f$)**
Reviewer $\f$ was the only reviewer who assigned our work a negative recommendation.
Their main concern relates to our definition of sycophancy, arguing a model can only be sycophantic in cases where it knows the answer and chooses to say the opposite. Instead, we argue that sycophancy is the tendency to align its responses with the user’s statement, regardless of whether the model knows the correct answer. We believe our definition is more accurate for the following reasons:
- Our definition of sycophancy is consistent with prior literature.
- We now also verified that for a sample $P$ in our benchmark, a model will attempt to prove $\neg P$, a factually correct statement, in the vast majority of cases (97.7%). Therefore, if a model also attempts to prove $P$ when asked to do so, the model’s output depends on the user's question, which constitutes sycophancy.
- Our framing is more relevant and realistic in actual use-cases. Users do not know whether the model has the inherent capability of solving the problem. Instead, they solely see whether the model attempted to prove the statement. Therefore, the numbers presented in our benchmark give a more direct indication how sycophancy affects user experience.
- All other concerns raised by reviewers have also been addressed in our rebuttal, we refer the AC to the rebuttal for more information. All requested changes have already been incorporated into the updated manuscript.

# Additional Experiments
In response to the reviewers, we also included several additional experiments, further strengthening our contribution:
- **Statistical Analysis**: We included a statistical analysis of all our results in the paper.
- **Prompt Variations**: We evaluate several additional prompt‑based mitigation strategies and show their effects on sycophancy.
- **Domain-Specific Breakdown**: We provided a detailed per-domain breakdown of our results.
- **Contamination Detection**: We performed a contamination detection experiment to support our claim that our benchmark is not contaminated.
- **Judge Validation**: We further expanded our validation of the LLM judge to show its high accuracy and reliability.

Best Regards,

The Authors

---

### Meta-Review · Area_Chair_HXcA · 2026-01-07

**Summary:**

Here are some of the principal reviewers' concerns

 - Construct validity. The benchmark’s “sycophancy” rate plausibly conflates alignment-driven conformity with mathematical incompetence, and the rebuttal does not implement the capability-filtering / identification needed to separate them. (FxiE)
 - Reviewer FxiE's another concern is downstream interpretability, because of the above mentioned confound, analyses like “harder to more sycophancy,” proof-vs-answer comparisons, self-sycophancy, and mitigation effects remain hard to interpret as alignment findings rather than competence effects. (FxiE)
 - Evaluator dependence. The core metric relies on LLM-as-judge, added cross-judge checks help, but concerns about residual judge error and limited human auditing (esp. on edge cases) remain. (evjq, XyWR, bA42)
 - Contamination and scale limits. “Recency” + a small probing subset do not conclusively rule out pretraining leakage, and n=504 constrains fine-grained, difficulty/domain-stratified conclusions. (evjq, XyWR)
- Dataset realism and generalization. LLM-generated perturbations may introduce benchmark-specific artifacts, and the scope (high-school/undergrad, natural language) leaves unclear transfer to research math or formal-proof settings. (XyWR)

**Reviewer Concerns:**

Here are some of principal reviewers' concerns that remained only partially addressed.

 - Core construct validity remains disputed. The rebuttal does not isolate “alignment sycophancy” from “can’t tell it’s false,” so the headline sycophancy rates still plausibly mix competence and conformity rather than identifying one cleanly. (FxiE)
 - Interpretability of all follow-on claims remains fragile. If the main metric is confounded, then difficulty trends, proof-vs-answer gaps, self-sycophancy, and mitigation deltas are still not clearly attributable to alignment vs ability. (FxiE)
 - Evaluation still hinges on LLM judging with limited human audit. Cross-family judging reduces family-bias concerns, but the overall dependence on an automated judge, and relatively small human-labeled set, leaves residual risk about edge-case labeling and error modes. (evjq, XyWR, bA42)
 - Contamination verification is still thin. Recency plus a small masked-word probe does not conclusively rule out leakage/near-duplicates, the rebuttal argues practicality but doesn’t provide stronger quantitative guarantees. (evjq)
 - Scope/generalization remains limited. The benchmark is natural-language and mostly HS/undergrad competition math, the rebuttal argues extensibility but does not empirically show transfer to research-level or formal-proof settings. (XyWR)

**Reviewer Scores:**

The  reviewer FxiE would have most probably kept the negative score. Some other reviewers  concerns remained only partially addressed, so other reviewers could have probably kept their scores.

---

### Decision · Program_Chairs · 2026-01-26

Reject